# Embodied Lifelong Learning for
# Task and Motion Planning

**Jorge Mendez-Mendez**
MIT CSAIL
jmendez@csail.mit.edu

**Leslie Pack Kaelbling**
MIT CSAIL
lpk@csail.mit.edu

**Tomás Lozano-Pérez**
MIT CSAIL
tlp@csail.mit.edu

**Abstract:** A robot deployed in a home over long stretches of time faces a true lifelong learning problem. As it seeks to provide assistance to its users, the robot should leverage any accumulated experience to improve its own knowledge and proficiency. We formalize this setting with a novel formulation of lifelong learning for task and motion planning (TAMP), which endows our learner with the compositionality of TAMP systems. Exploiting the modularity of TAMP, we develop a mixture of generative models that produces candidate continuous parameters for a planner. Whereas most existing lifelong learning approaches determine a priori how data is shared across various models, our approach learns shared and non-shared models and determines which to use online during planning based on auxiliary tasks that serve as a proxy for each model's understanding of a state. Our method exhibits substantial improvements (over time and compared to baselines) in planning success on 2D and BEHAVIOR domains [1].

**Keywords:** task and motion planning, lifelong learning, generative models

## 1   Introduction

Consider a home assistant robot operating over a lifetime in a home. The robot initially comes equipped with a number of basic capabilities for planning and control that enable it to execute certain actions, such as NAVIGATETO(object) and GRASP(object). The robot's user expects it to leverage these abilities to immediately assist with house chores, and to become increasingly capable over time, adapting to the types of problems that arise in its new home environment. This setting evokes a novel lifelong learning formulation for embodied intelligence that necessitates learning *in the field*, which we formalize and address in this work (Figures 1 and A.1). Notably, we forgo any artificial separation between training and testing: the robot is continually asked to solve problems to the best of its abilities, and it is free to use any data it collects to improve its knowledge for future use.

One promising tool for tackling this lifelong learning challenge is planning. Planning models comprise relatively independent prediction modules, which permits learning disentangled knowledge and reusing it compositionally. Similar forms of modularity have been shown to provide substantial leverage for training models continually by targeting the learning to individual modules, composing existing modules into new solutions, and adding new modules over time [2].

In this paper, we focus on learning generative models to address the most difficult aspect of task and motion planning (TAMP) [3]: finding continuous parameters (grasps, poses, paths) that guarantee the success of a high-level plan. Our approach learns to generate samples that lead to problem completion, implicitly considering the effect of the current sample on future samples' success. Compared to other applications of generative models, one peculiarity of the lifelong sampler learning problem is that it is inherently multitask, since TAMP systems often apply similar basic skills (e.g., GRASP) to varied object types (e.g., ball, box). At two extremes of possible approaches to this multitask problem, we could learn an independent, specialized sampler for each object type, or we could learn a single, generic sampler across all types. Intuitively, specialized models are likely to yield better

7th Conference on Robot Learning (CoRL 2023), Atlanta, USA.

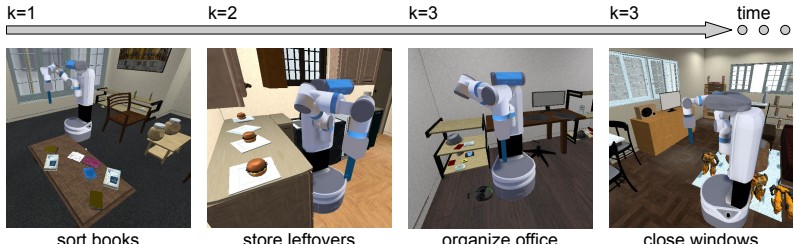

Figure 1: The learning robot will face a sequence of diverse TAMP problems in a true lifelong setting. It will use its current models to solve each problem as efficiently as possible, and then use any collected data to improve those models for the future. Images captured from BEHAVIOR [1].

predictions, but they are trained on a smaller pool of data, making them potentially less robust. We propose a hybrid solution that learns both general and specialized samplers, and generates samples by determining online which model is most appropriate given a state. To do so, we pair each sampler with an auxiliary predictor. When drawing samples for solving a problem, the *measured* error on the auxiliary tasks serves as a proxy for the sampler's accuracy at the given state, which we use to construct a mixture distribution that assigns more weight to lower-error samplers.

We evaluate the resulting TAMP system for its cumulative performance as it encounters a sequence of diverse problems. This realistic formulation measures how effective the robot is, in aggregate, at solving problems over its entire lifetime. This contrasts with standard (artificial) lifelong learning evaluations, which instead focus on the final performance over the sequence of problems used for training. Concretely, we evaluate our approach on problems from a 2D domain and from the BEHAVIOR benchmark [1], demonstrating substantial improvements in planning performance.

## 2 The Lifelong Sampler Learning Problem

Our formulation of lifelong sampler learning emphasizes a realistic setting in which a robot is deployed in an environment and, from that moment on, is continually evaluated for its ability to solve TAMP problems. This section first formalizes the problem in terms of general TAMP systems, and next discusses how it applies specifically to search-then-sample (SeSamE) bilevel planning strategies.

### 2.1 Learning Over a Sequence of TAMP Problems

The lifelong learning robot faces a continual stream of planning problems $\tau^{(1)}, \tau^{(2)}, \ldots$. Each problem $\tau^{(k)} = \langle s_{\text{init}}^{(k)}, g^{(k)} \rangle \sim \mathcal{D}^{(k)}$ is defined by an initial state $s_{\text{init}}^{(k)}$ and a goal $g^{(k)}$, drawn sequentially from some problem distribution $\mathcal{D}^{(k)}$, which is potentially nonstationary (as indicated by the superscript $k$). We assume that the robot has access to a sound and probabilistically complete planner; while the planner is guaranteed to solve any problem given sufficient computation time, we expect many problems to be infeasible within a reasonable time. In consequence, the robot should leverage any available data to focus its planning process on choices that are likely to be successful.

Like in the real world, there are no distinct training and evaluation stages. Instead, at each time $k$, the robot seeks to construct a plan for a new problem $\tau^{(k)}$. We record whether the robot succeeds at the problem within a bounded number of samples $B$, and how many samples it attempts. The robot may use any experience it collects during the planning attempt as training data to improve its models. We measure performance as the cumulative number of problems solved within a given number of attempted samples. A system that learns efficiently from the data available so far will solve new problems more quickly, gaining even more data from successful plans, in a virtuous cycle.

One attribute of this formulation is that the agent is never evaluated on any previous problem—after time $k$, problem $\tau^{(k)}$ is never exactly encountered again. This departs from existing lifelong learning formulations, which evaluate the agent on previous problems $\tau^{(1)}, \ldots, \tau^{(k-1)}$ to measure forgetting. Yet, avoiding forgetting still contributes to attaining good performance in our setting. Since we

evaluate the robot continually on *new* problems, it must generalize from prior experience. If the robot forgets past knowledge that remains relevant and overfits to the latest problems, then it will never improve over the distribution of problems that it may face in the future. Even in nonstationary cases, where later problems differ substantially from earlier ones, retaining knowledge of past instances may improve generalization to future problems; we demonstrate this empirically in Section 5.3.

In order for the robot to generalize in this fashion, the problem distribution $\mathcal{D}^{(k)}$ must consist of problems that have some common underlying structure, enabling samplers to be shared across problems to serve as the medium through which prior experience informs future planning.

## 2.2 SeSamE Bilevel Planning

One strategy to solve TAMP problems is to separate the search into two disentangled levels to perform bilevel planning [4]. We adopt this strategy and formalize the bilevel planning in a variant of PDDL [5] augmented with continuous parameters. The robot will be deployed in a world $\mathcal{W} = \langle \Theta, \mathcal{R}, \mathcal{S}, \mathcal{O}, \mathcal{A} \rangle$. Each object $e$ of type $\theta_e \in \Theta$ is described by a feature vector $\boldsymbol{x}$. A state $s \in \mathcal{S}$ is characterized by the features $\boldsymbol{x}$ describing all objects. The pos-

---

**Algorithm 1** SeSamE$(\tau, N, M)$

skeleton_gen $\leftarrow$ discreteSearchSolutionGen$(\tau)$
**for** $j = 1, \ldots, N$ **:** $\quad\quad\quad$ ▷ Loop over skeletons
$\quad$ skel $\leftarrow$ skeleton_gen.next() $\quad$ ▷ Next skeleton
$\quad$ cnt $\leftarrow$ zeros(len(skel)) $\quad\quad$ ▷ Step sample counter
$\quad$ **for** $i = 1, \ldots,$ len(skel) **:** $\quad\quad$ ▷ Loop over steps
$\quad\quad$ $\phi[i] \leftarrow$ sample(skel[$i$], $s[i]$); cnt[$i$]++
$\quad\quad$ $s[i+1] \leftarrow$ simulate($s[i]$, skel[$i$], $\phi[i]$)
$\quad\quad$ **if** not valid($s[i+1]$) **:**
$\quad\quad\quad$ **while** cnt[i] $= M$ **:** cnt[$i$--] $\leftarrow 0$ $\quad$ ▷ Backtrack
$\quad\quad\quad$ $i$-- $\quad$ ▷ Set $i$ so latest cnt[$i$] $< M$ is re-sampled
$\quad\quad\quad$ **if** $i = 0$ **:** break $\quad\quad\quad$ ▷ Try new skeleton
$\quad$ **if** $s[i] \in \tau.$goal **:** **return** skel, $\phi$

---

itive predicates $r \in \mathcal{R}$ on $s$ produce an abstract state $s^\uparrow$ (e.g., IN(ball, box) $\wedge$ ON(box, floor)). An abstract action $a^\uparrow = (o, \mathcal{C})$ is an operator $o \in \mathcal{O}$, with predicate-level preconditions and effects, augmented with a parameterized controller $\mathcal{C}^{a^\uparrow}_{\phi, e}$. The parameters of the controller are hybrid: $e$ are discrete typed parameters that specify which objects to apply the controller to, while $\phi$ are continuous parameters that dictate how the action is applied. For example, a NAVIGATETO action applied to $e =$ table could take continuous parameters $\phi = (\Delta_x, \Delta_y)$ that specify the target coordinates, relative to the table. An action $a \in \mathcal{A}$ is the execution of a controller with given parameters. We assume that all symbolic elements of $\mathcal{W}$ are given, and focus on determining the continuous parameters $\phi$ that result in actions that lead to complete plans. A planning problem $\tau = \langle s_{\text{init}}, g \rangle$ is given by a continuous-level initial state $s_{\text{init}}$ and a predicate-level goal $g$. A satisficing plan $\pi$ is a sequence of actions $a_1, \ldots, a_j$ which move the robot from $s_{\text{init}}$ to an abstrct state where $s^\uparrow \subseteq g$.

The SeSamE strategy (Algorithm 1) first obtains a skeleton plan at the discrete level using standard planning techniques (e.g., $A^*$), and subsequently refines the skeleton into a low-level plan via sampling. While generating samples is inexpensive, the simulation step that evaluates each sample can be quite expensive (e.g., due to inverse kinematics, collision checking, or path planning). Reducing the number of samples required to obtain a satisficing plan, and hence the number of calls to the simulator, can substantially improve the efficiency of the TAMP system, vastly increasing the space of problems that can be solved in a tractable amount of time.

One challenge in this sampling-based formulation is specifying a distribution that 1) covers the space of plausible solutions and 2) generates only promising candidates. Learning such a distribution from data, as we do in this work, would constitute a major step toward constructing an effective TAMP system.

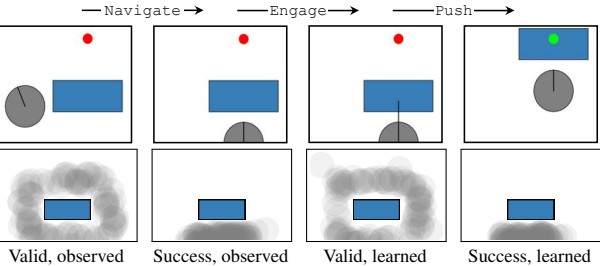

Figure 2: Sample distributions optimized for TAMP problem completion. Top: a 2D robot tasked with pushing a block upward. Bottom: distribution of navigation parameters that achieve reachability (valid) and that solve the problem (success), generated by an expert (observed) and learned by a diffusion model (learned). Samples that consider global success yield a distribution over promising (and not merely valid) actions. Diffusion models represent the observed (success or valid) distributions well.

### 2.3 Samplers as Generative Models of Plausible Candidate Action Parameters

The sampler for each abstract action $a^\uparrow$ should generate action parameters from a distribution conditioned on the current state: $\phi \sim p(\cdot \mid s, a^\uparrow)$. This distribution should capture the whole space of action parameters that, given the current state, may lead to successful execution of an overall plan. To illustrate this point, consider the block-pushing domain in Figure 2 (see Section 5.1 for details). A naïve navigation sampler could bring the robot within reach of the block, but a more intelligent sampler would only consider navigating to the bottom of the block to enable pushing upward.

To learn a generative model that meets these criteria, we require access to a pool of paired states and action parameters $(s, \phi)$ that is representative of the choices that eventually (after successfully sampling all subsequent actions in a skeleton plan) lead to a satisficing plan. Critically, the action parameters that succeed for a given abstract action $p_{a^\uparrow}$ depend on the samplers for the remaining abstract actions, and as such the data must be collected jointly to ensure that the learned distributions are compatible with each other. The lifelong problem formulation of Section 2.1 satisfies this criterion.

## 3 Nested Models for Sparse Data

In our lifelong learning setting, the robot should use whatever data it has to make the best predictions possible. To handle the sparse-data setting that inevitably occurs early in the robot's experience, we formulate a solution that nests predictors of different generality.

In order to train neural generative models as the samplers for our TAMP system, we construct a vector representation of the conditioning variables. The sampling distribution for each abstract action $a^\uparrow$ is given by $p(\phi \mid s, C_{\phi,e}^{a^\uparrow})$. We first represent the state $s$ in terms of the continuous low-level features of the objects involved in the action, $x_o$, following prior work [4]. A trivial second step is to train separate models for each form of controller, $p_C$, since different controllers generally have different parameterizations, and so there are no commonalities across their sample distributions. However, each such controller may act on objects of different natures in ways that require substantially different parameters: this distinction is encoded in the discrete variable corresponding to the object types $\theta_o$.

Thus, we would like the predictions of our model to be specialized to each individual object type. For this, we can consider the following two strategies (omitting the $C$ subscript for clarity):
1. Learn a single model $p(\phi \mid x_o, \theta_o)$ using, for example, a one-hot encoding of the types. This would enable the learning of a single, shared latent representation of $x_o$ in the neural net model. However, in cases where the amount of data is small, it is more likely to simply overfit.
2. Learn a separate, simpler, model $p_\ell(\phi \mid x_o)$ for each discrete value $\ell$ of $\theta_o$.

However, early on, when the robot has observed very little data, we may prefer an even more aggressive strategy that pools data over all values of $\theta_o$, and learns a single simple model $p(\phi \mid x_o)$. This model will not be highly specific or accurate, but it may learn more quickly to generate reasonable suggestions, due to pooled data. It is also important to observe that, in the lifelong setting, we may have an asymmetric distribution of experience with samplers for different object types, so that some specific models would have substantially more training data available than others, in a way that cannot be predicted in advance and that will change over time.

For all these reasons, we propose a *nested* approach: train both a generic sampler $p(\phi \mid x_o)$ and a collection of specialized samplers $p_\ell(\phi \mid x_o)$ for all values $\ell$ of $\theta_o$. Then, we decide *online*, given an input with $\theta_o = \ell$, based on an assessment of prediction reliability, how much to value predictions from the generic versus the specialized samplers, and actually sample from a mixture distribution.

To construct this assessment of the reliability of each generative model, we construct an auxiliary training task to predict a variable $z$, and augment each training example, so we have $(s, \phi, z)$. Critically, $z$ must be a value that can be directly measured by the robot, so that the error between its learned predictor $f(s, \phi)$ and the actual observed $z$ can serve as a measure of how well trained $p(\phi \mid s)$ is in the part of the input space near $s$. So, in parallel with training $p(\phi \mid s)$, we will train $f(s, \phi)$, and in parallel with training each $p_\ell(\phi \mid s)$, we will train a *separate* specialized $f_\ell(s, \phi)$.

At planning time, if the robot must draw a sample in state $s$ for the action applied to objects with types $\ell = \theta_o$, it will use the following mixture distribution:

$$p_{\text{mix}}(\phi \mid s, z) = \frac{\rho(f(s,\phi), z)p(\phi \mid s) + \rho(f_\ell(s,\phi), z)p_\ell(\phi \mid s)}{\rho(f(s,\phi), z) + \rho(f_\ell(s,\phi), z)} \quad , \tag{1}$$

where $\rho(f_\ell(s,\phi), z)$ and $\rho(f(s,\phi), z) \in \mathbb{R}^+$ are *reliability measures* of the specialized and generic models at $(s, \phi)$, constructed by comparing their predictions to observed $z$ (for example, the inverse of the squared prediction error). Note that, since the auxiliary signals $z$ depend on the action parameters $\phi$, the sampling process must draw samples from the two mixture components, use them to compute the reliability measures, and then choose between the samples based on the resulting mixture weights.

**Implementation**  In our implementation, we use simple observable geometric properties of the world state as $z$ values, restricted to the objects $e$ that parameterize the action. Although they would not suffice for selecting good samples $\phi$, since they do not consider the effect of $\phi$ on future steps nor the interactions with objects outside of $e$—like a complete simulator would—our ability to predict them accurately is a signifier that the accompanying model has obtained sufficient data in the neighborhood of $(s, \phi)$ to consider $\phi$ a good prediction. As an example, consider the NAVIGATETO(block) action from Figure 2. A useful set of auxiliary signals for this case may be: the orientation of the robot facing the block, the distance to the center of the block, the distance to the nearest point on the block's boundaries, and the relative coordinates of this point and the robot's center in the block's coordinate frame. Combined, these signals contain abundant information about the effects of the action, including those relevant for reachability and collision avoidance with the block. A model that learns to map a $(s, \phi)$ pair to accurate predictions of all these signals has likely observed sufficiently similar training pairs to generalize, and is therefore likely to generate high-quality samples $\phi$. See Appendices B.2.2 and B.3.2 for the exact auxiliary signals used in our experiments.

## 4 Diffusion Models for Parameter Sampling

We use diffusion models to represent our learned samplers, due to their stability of training and their ability to model complex distributions. A diffusion model transforms Gaussian noise into a distribution over the sample space. To do so, it generates training data by following a forward diffusion process, which progressively adds Gaussian noise to observed training samples, and trains a reverse diffusion process that gradually denoises Gaussian noise to produce a sample from the learned distribution [6]. Concretely, when learning TAMP samplers, the forward diffusion process is given by $q(\phi_{0:T} \mid s) = q(\phi_0 \mid s) \prod_{t=1}^{T} q(\phi_t \mid \phi_{t-1})$, where each step $q(\phi_t \mid \phi_{t-1})$ adds Gaussian noise to $\phi_{t-1}$ and $q(\phi_0 \mid s)$ denotes the observed distribution of successful action parameters $\phi$. The reverse process is the generative model parameterized by $\psi$, and is similarly defined as a Markov chain: $p_\psi(\phi_{T:0} \mid s) = p(\phi_T) \prod_{t=T}^{1} p_\psi(\phi_{t-1} \mid \phi_t, s)$, where $p(\phi_T)$ is a standard Gaussian prior. Each step $p_\psi(\phi_{t-1} \mid \phi_t, s) = \mathcal{N}(\phi_t - \epsilon_\psi(\phi_t, s, t), \sigma^2 \boldsymbol{I})$ estimates the mean of a Gaussian by subtracting predicted noise given a noisy version of the action parameters, the state, and the time step. Once trained, the planner can sample from the model by simulating the reverse process $p_\psi(\phi_{T:0} \mid s)$.

Separating the learning process into various diffusion models, each in charge of representing a specific distribution, as described in Section 3, reduces the difficulty of lifelong training as compared to training a single model to fit all distributions. Even so, each such diffusion model still requires continual training: a specialized sampler $p_\ell(\phi \mid \boldsymbol{x}_o)$ receives additional data as the robot uses it to attempt to solve new problems, and a generic sampler $p(\phi \mid \boldsymbol{x}_o)$ additionally receives data from new object types as the robot encounters them. Given a previous pool of data $\mathcal{Z}_{\text{old}}$ and a newly acquired set of data points $\mathcal{Z}_{\text{new}}$, we consider the following training schemes:

- **Finetuning**  Starting from the previous model, train on $\mathcal{Z}_{\text{new}}$, likely forgetting previous knowledge.
- **Retraining**  Train a new model on $\mathcal{Z}_{\text{new}}$ and $\mathcal{Z}_{\text{old}}$ jointly. In practice, this method has been shown to outperform all true continual training approaches, at the cost of high computational expense.
- **Replay**  Starting from the previous model, train over a *balanced* sampling of data from $\mathcal{Z}_{\text{new}}$ and $\mathcal{Z}_{\text{old}}$ [7]. While more advanced strategies are possible, we found that this simple method matched the retraining performance with only $10\%$ of the training epochs used for retraining.

# 5 Experimental Evaluation

Our initial experiments sought to validate that diffusion models can learn useful TAMP samplers and that the mixture distribution proposed in Section 3 achieves good performance in an offline setting. The results of these experiments set up our main evaluations over lifelong sequences of 2D and BEHAV-IOR domains. Additional details are provided in Appendix B.

## 5.1 Visualizing Sampling Distributions

To illustrate the usefulness of our approach to learning samplers for TAMP, we created two short-horizon, 2D domains that permit us to visualize the learned distributions. The first domain (Figure 2, top) requires a robot to push a block upward. We focus on the navigation action, parameterized by the 2D coordinates, which succeeds if the block becomes reachable. However, the block can only be pushed from the bottom, so most valid actions do not lead to success. The second domain (Figure 3, top) requires placing two blocks in an L-shaped container, where the first block fits in any of the two sections, but the second block only fits in the long section. The blocks are placed individually by selecting their 2D pose, but not all valid placements of the first block enable problem success.

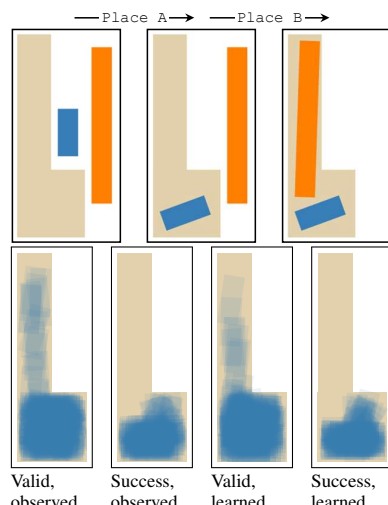

Figure 3: Top: a planner must fit two blocks in a container that restricts the longer block's placement. Bottom: distribution of parameters for achieving placement of the small block (valid) or solving the problem (success), generated by an expert (observed) and a diffusion model (learned). The distributions match.

We collected two data sets for each domain: one with all valid actions and one with only actions that lead to problem success. We then trained a diffusion-based sampler on each data set. Figures 2 and 3 (bottom) show the observed and learned distributions, demonstrating that optimizing for problem success yields distributions that consider the long-term effects of actions, even if future constraints are not observable by the sampler. The learned distributions match the observed distributions well.

## 5.2 Learning Samplers from Fixed Data on 2D Domains

We next studied the impact of using our nested models in solving a variety of TAMP problems in an offline setting. We created five 2D robotics domains, each of which requires navigating, picking, and placing different objects in a target container. The domains vary in the object size ranges, graspable regions, and container size ranges and shapes (see Appendix B.2.1 for additional details). We collected demonstration data for solving $K$ problems from each domain, trained the samplers on the fixed data sets, and evaluated their efficiency in solving $K' = 50$ unseen problems. We repeated the evaluation over 10 trials with various random seeds controlling the test problem generation.

Our experiments considered the following (primarily) diffusion-based sampler learning methods:
- Specialized: separate for each typed abstract action
- (CD) generic: shared across actions with a common controller within (across) domains
- (CD) mixture: our approach mixing specialized and (CD) generic samplers
- NSRTs: The sampler learning component of Chitnis et al.'s [4] work—a non-diffusion-based model

Figure 4 shows average results for varying numbers of training problems $K$ (standard errors in Appendix C). As expected, all methods become more efficient at generating good samples as they are trained on increasing amounts of data. In particular, the specialized samplers (which observe the least amount of data) are the most inefficient when trained on $K = 50$ problems, but become as efficient as generic samplers upon training on $K = 50{,}000$ problems. Our mixtures of nested models are substantially more efficient than alternative methods when trained with little data. Appendix D analyzes various mechanisms for selecting between specialized and generic samplers, demonstrating that using geometric auxiliary signals as a proxy for sampler generalization is a strong choice.

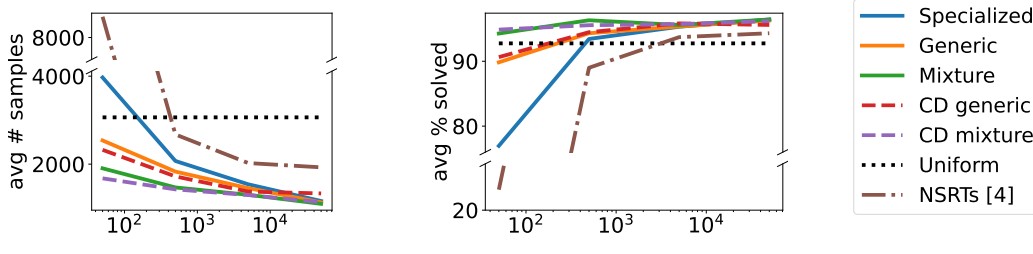

(a) Number of samples per solved problem      (b) Number of problems solved

Figure 4: Results of learning diffusion models as TAMP samplers from offline data. Given sufficient data, all samplers solve the majority of problems efficiently. In the small-data regime (note the log-scale of the x-axis), sharing data across samplers improves sampling efficiency. The mixture distributions learned either individually on each domain (Mixture) or across all domains (CD mixture) are best, thanks to their ability to automatically select generic or specialized samplers during planning.

## 5.3 Lifelong Evaluation on 2D Domains

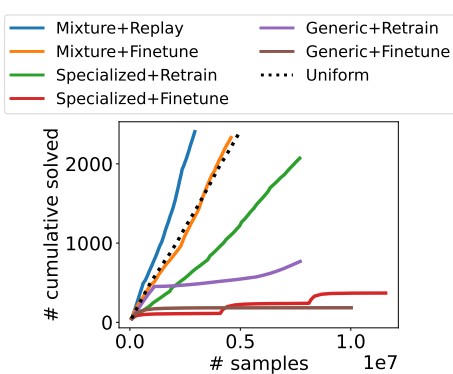

Our lifelong evaluation presented the agent with a sequence of problems from each domain in turn (first domain 1, then domain 2...), imposing a nonstationary distribution. The agent attempted to solve each problem given its current sampler, and subsequently used the collected data for additional training. There was no separate test phase: the agent's performance was assessed over its attempts to solve the problems in the sequence (before training on them). We updated the models of each method using the retraining, finetuning, and replay strategies of Section 4.

Figure 5: Lifelong learning results on 2D domains (avg. over 10 seeds). The mixture distributions are vastly superior. Finetuning the models directly fails due to forgetting, especially with generic and specialized samplers.

Figure 5 shows the cumulative number of problems solved against the number of generated samples. We used retraining for specialized and generic samplers as an approximate upper bound on their performance. Our mixture sampler is substantially more efficient than baselines, and is the only approach that outperforms a naïve uniform sampler. In the lifelong setting, agents use their current samplers to collect data, which explains why even the retraining version of the generic baseline fails. The agent uses the model trained on previous types to generate samples for novel types; early on, those samplers are overfit to the small set of initial types, causing them to fail to solve any problems and preventing the robot from acquiring useful data for subsequent updates. Specialized samplers work reasonably well, but are slowed down by their inability to quickly generate samples when trained over little data. The finetuning approaches, which are known to suffer from forgetting, perform noticeably worse, demonstrating the need to retain knowledge throughout the prlblem sequence.

## 5.4 Lifelong Evaluation on BEHAVIOR

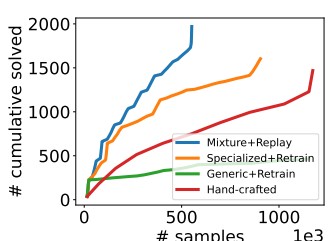

We next applied the evaluation protocol of Section 5.3 to 10 families of BEHAVIOR problems (see Appendix B.3 for details). Figure 6 demonstrates that our method enables continual learning over this more complex and realistic domain. Notably, the agent improves even over the hand-crafted samplers that we provided as a starting point, demonstrating the usefulness of our approach even when engineered solutions are already in place.

Figure 6: Lifelong learning results on BEHAVIOR. Our lifelong learner progressively improves over its initial (handcrafted) samplers, becoming increasingly better at solving diverse problems.

# 6   Related Work

**Lifelong learning**    Recent literature on lifelong learning has primarily focused on avoiding catastrophic forgetting [8] in supervised settings [9, 10, 11, 12, 13]. Various techniques achieve this by replaying past data [14, 15, 7], which we adopt. Our focus is on enabling robots to become increasingly capable over time via compositionality. Few works have studied compositionality in lifelong supervised [16, 17, 18] and reinforced [19, 20, 21] domains. Prior lifelong learning work considers training an agent over a sequence of machine learning tasks and subsequently evaluating the system over those same tasks. We propose a more natural evaluation setting without train/test splits, where the agent continually seeks to solve TAMP problems and uses the data from those problems to learn. While seemingly this formulation does not require forgetting avoidance (since previous problems are never encountered again), prior work has suggested that knowledge retention is useful for generalization to *future* problems [2]. Our results in Sections 5.3 and 5.4 exemplify this notion.

**Learning for TAMP**    Numerous recent methods seek to broaden the capabilities of TAMP systems beyond engineered solutions via learning. A majority of such methods focus on symbolic aspects of the problem: given a partial symbolic description of a domain, use demonstration data to fill the gaps in symbolic space to enable solving additional problems. This can take the form of learning operators (i.e., action abstractions) [22, 23, 4] or predicates (i.e., state abstractions) [24, 25, 26]. Given the difficulty of mapping abstract actions to continuous robot actions, our focus is on learning at the continuous level, specifically in the form of samplers. The most closely related approach, which learns samplers for SeSamE planners, considers a very simple class of regression samplers with a learned rejection classifier [4]. Other methods for learning TAMP samplers use more powerful generative models, but not diffusion models [27, 28, 29, 30, 31]. None of these prior works study the underlying multitask problem, nor do they operate in a lifelong setting as our method does. Other (less) related work uses reinforcement learning to bridge between the discrete and continuous actions [32, 33] or automatically decomposes motion plans into discrete components [34].

# 7   Conclusion and Limitations

We proposed a novel formulation of the embodied lifelong learning problem for TAMP systems, which emphasizes realistic evaluation of the robot as it attempts to solve problems over its lifetime. Our solution approach learns a mixture of nested generative models, assigning higher weight to models that attain low error on auxiliary prediction tasks. Our experiments on 2D and BEHAVIOR domains demonstrate the ability of our approach to acquire knowledge over a lifetime of planning.

**Limitations of chosen TAMP methods**    We adopt a SeSamE strategy for planning, which requires a faithful environment simulator. In environments that are difficult to simulate (e.g., because they contain many objects), we would prefer a different TAMP method; we leave this problem for future work. Moreover, our samplers are conditioned only on the objects involved in the action. Conditioning on all other objects or on the plan skeleton could generate better samples for problem completion.

**Limitations of the lifelong setting and approach**    One direction to improve the learning is to develop better exploration strategies. Our method directly uses the learned models to generate samples; trading off exploration and exploitation could yield additional benefits. Additionally, our lifelong training replays all past data. In our experiments, this represents only a small portion of the computation, but it would become infeasible in longer deployments, so subsampling strategies—which our approach can trivially adopt—should be employed. Our lifelong learning formulation builds upon operations that the robot can already execute. It would also be valuable to study the autonomous learning of *new* operations. We encourage future research in this direction.

**Limitations of the experimental setting**    We measure number of samples as a proxy for planning time. In practice, there is a trade-off between simulation cost and sampling cost; expensive diffusion models might lead to higher sampling cost than simulation cost in some cases. Relatedly, we have not yet evaluated our approach on physical robots. The facts that our method builds upon a working TAMP system and that it is highly data efficient suggests that hardware deployment is probable.

**Acknowledgments**

We thank Nishanth Kumar, Willie McClinton, and Kathryn Le for developing and granting us access to the base TAMP system for BEHAVIOR. We also thank Yilun Du for initial discussions and guidance on diffusion models. The research of J. Mendez-Mendez is funded by an MIT-IBM Distinguished Postdoctoral Fellowship. We gratefully acknowledge support from NSF grant 2214177; from AFOSR grant FA9550-22-1-0249; from ONR MURI grant N00014-22-1-2740; from ARO grant W911NF-23-1-0034; from the MIT-IBM Watson AI Lab; from the MIT Quest for Intelligence; and from the Boston Dynamics Artificial Intelligence Institute.

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

# Appendices to
# "Embodied Lifelong Learning for
# Task and Motion Planning"

**Anonymous Author(s)**

## A    Visual Depiction of the Nested Lifelong Sampler Learning Approach

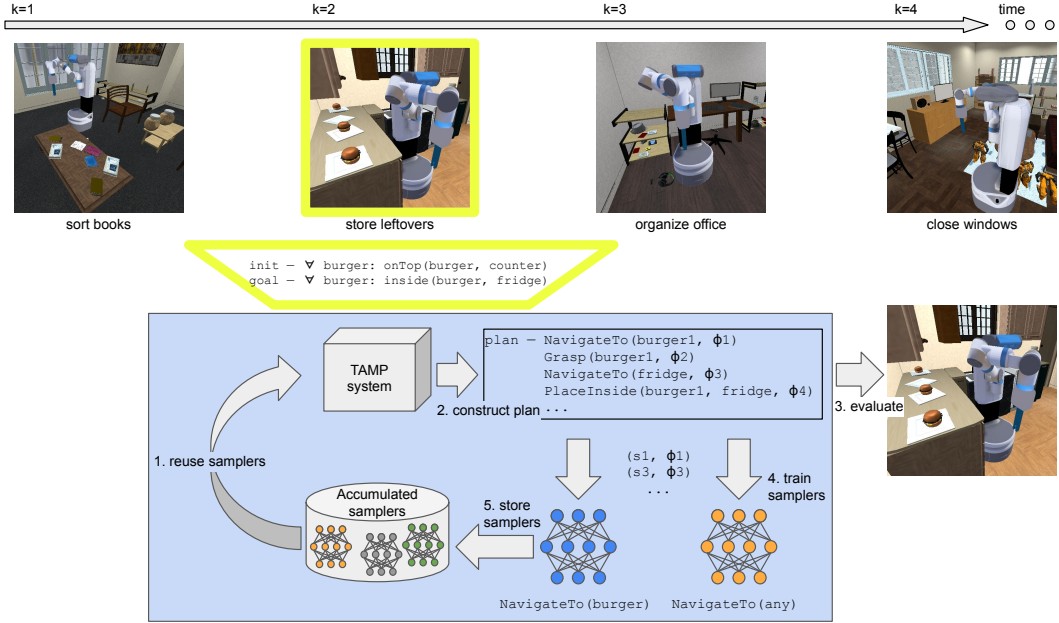

Figure A.1: The learning robot will face a sequence of diverse TAMP problems in a true lifelong setting. It will use its current models to solve each problem as efficiently as possible, and then use any collected data to improve those models for the future. Images captured from BEHAVIOR [1].

## B    Experimental Details

This section provides additional details about the experiments in Section 5 in the main paper.

### B.1    Network Architectures and Diffusion Models

All network architectures used throughout our work were simple multi-layer perceptrons with two hidden layers of 256 nodes each. For our nested models, the hidden layers were shared between the generative model $p_{\psi}$ and the auxiliary predictor $f$, and only the output layer was separate—note that there was no sharing of parameters between specialized $p_{\ell}$ and generic $p$ samplers. All training proceeded for 1,000 epochs over the training data in mini-batches of size 512 (when that many samples were available, and a single batch otherwise), using an Adam optimizer with the default hyperparameters of PyTorch, including a learning rate of $10^{-3}$. The one exception was the replay method for lifelong training, which used 100 epochs of training during model adaptation.

To train the diffusion models, for each point $(s, \phi)$, a time step $t \in [1, T]$ was randomly chosen for $T = 100$, and a sample was drawn from the forward process $\phi_t = \phi\sqrt{\bar{\alpha}_t} + \epsilon\sqrt{1 - \bar{\alpha}_t}$, where $\epsilon \sim \mathcal{N}(\mathbf{0}, \mathbf{I})$ is standard Gaussian noise and $\bar{\alpha}_t$ is a scaling constant obtained by expanding the expression of $q(\cdot)$. The loss function then measured how closely $\epsilon_{\psi}(\phi_t, \boldsymbol{x}, t)$ approximated the true noise $\epsilon$: $\mathcal{L}(s, \phi, t, \epsilon) = \|\epsilon_{\psi}(\phi_t, \boldsymbol{x}, t) - \epsilon\|$. Once trained, the planner sampled from the model by simulating the reverse process $p_{\psi}(\phi_{T:0} \mid s)$.

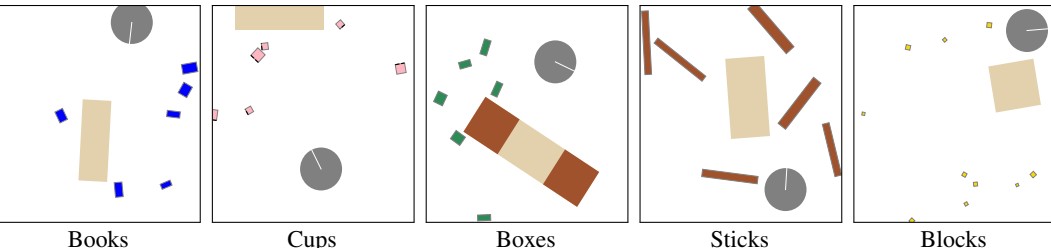

Figure B.2: 2D domains used to evaluate our sampler learning approaches. The objects in each domain have properties that ensure that samplers must generate diverse candidate action parameters to solve the problems.

To process the input composed of three vectors $(\phi_t, \boldsymbol{x}, t)$, the time index $t$ was first processed using sinusoidal positional embeddings [6] of the same dimension as $\boldsymbol{x}$. Then, the three vectors were concatenated into a single input to the network. Since the auxiliary predictor $f$ shared the same base layers of the network, we used $t = 0$ as a constant input to $f$.

All inputs and outputs were scaled to $[0, 1]$, except when the range of the variable was less than $1$, in which case the given variable was shifted to $0$, but not rescaled.

## B.2    2D Domain Experimental Setting

We now provide the details of our evaluations on the 2D domains.

### B.2.1    Domain Descriptions

We created five different 2D domains, specially crafted to require distinct sampling distributions across objects. Figure B.2 depicts the simulated domains. As described in Section 2.2 in the main paper, each problem within a domain is a sampled initial state and a goal. In this case, all goals are of the form "place all objects in the container."

- **Books**  Rectangular books of sides $w_{\mathrm{book}} \in [0.5, 1], l_{\mathrm{book}} \in [1, 1.5]$ are scattered in a room and must be picked and placed on a rectangular shelf of sides $w_{\mathrm{shelf}} \in [2, 5], l_{\mathrm{shelf}} \in [5, 10]$.

- **Cups**  Square cups with sides $l_{\mathrm{cup}} \in [0.5, 1]$ must be picked *by the handle* (one specific side) and palced in a cupboard of sides $w_{\mathrm{cupboard}} \in [2, 5], l_{\mathrm{cupboard}} \in [5, 10]$. The cupboard is always against a wall.

- **Boxes**  Boxes of sides $w_{\mathrm{box}} \in [0.5, 1], l_{\mathrm{box}} \in [1, 1.5]$ must be placed *on pockets at the extreme ends* of a tray of width $w_{\mathrm{tray}} \in [3, 5]$ and length $l_{\mathrm{tray}} \in [11, 13]$.

- **Sitcks**   Long sticks of sides $w_{\mathrm{stick}} \in [0.5, 1], l_{\mathrm{stick}} \in [5, 6]$ in a container of sides $w_{\mathrm{container}} \in [3, 5], l_{\mathrm{container}} \in [7, 10]$.

- **Blocks**  Small square blocks of sides $l_{\mathrm{block}} \in [0.25, 0.5]$ must be place in a square bin of sides $l_{\mathrm{bin}} \in [4, 6]$. While previous problems contain $n \in [4, 5]$ objects, these require placing $n \in [9, 10]$ blocks.

The robot has three controllers that it can execute: NAVIGATETO(`object`), parameterized by the relative target coordinates normalized by the `object`'s size; PICK(`object`), parameterized by the length to extend the robot's gripper and the angle to hold the `object` at; and PLACE(`object`, `container`), parameterized by the gripper extension.

### B.2.2    Auxiliary Geometric Signals

This section describes the auxiliary signals we used to train our predictors $f$, and we later describe how those were used to construct the mixture distributions for our samplers. We used the following auxiliary signals for each form of controller:

- NAVIGATETO: distance to the nearest point between the robot and the object—in the case of the cup, this signal measured distance to the handle, and in the case of the tray, it measured distance to the nearest pocket; coordinates of the nearest point on the object's boundaries (in the object's frame); and target coordinates (in the world frame and in the object's frame).

- PICK: position of the gripper's point (in the world frame and in the object's frame).

- PLACE: center of mass of the object (in the world frame and in the container's frame).

Note that all these signals measure the *intended effects* of an action, but cannot measure the actual attained effect, which would require knowledge of all objects in the world. However, by measuring the agent's accuracy in predicting these signals, we can asses how well-trained it is in the neighboring region of the current state, and use that as a measure of how well its samples may generalize.

### B.2.3 Constructing the Mixture Distribution

In these 2D domains, we created mixture distributions over three mixture components: a generic sampler trained on all object types, a specialized sampler for each object type, and a fixed uniform sampler over the parameter space of the controller. We used the inverse of the root mean square error as the assessment of reliability, $\rho$. For this, we first computed (offline) the average prediction error for random guessing via simulation, and assigned this fixed error value to the uniform sampler. Then, we used this value to normalize prediction errors across the various signals.

### B.2.4 Lifelong Training Details

In the lifelong setting, upon facing a new problem, the agent used its mixture sampler to generate samples for any previously seen object type. For unknown types, the agent used a mixture over the generic and the uniform sampler with fixed weights of $0.5$. At the very beginning, the samplers were initialized with a uniform distribution over the parameter space.

### B.2.5 Evaluation Protocols

In the offline setting, we generated 50 test problems for each of 10 trials, with varying random seeds controlling the sizes of objects and their placements.

In the lifelong setting, each trial shuffled the order of the domains using the random seed, and presented the agent with a sequence of 500 problems from each domain. Instead of updating the models after each problem, which would render most updates very minor, we updated the models at intervals of 50 problems, resulting in a total of 10 model updates per domain.

In both settings, we used Fast-Downward as the skeleton generator, getting a single skeleton for each problem (i.e., $N = 1$) and setting the maximum total number of samples to $B = 10{,}000$. During search, a maximum of $M = 100$ samples were attempted at any given state before backtracking. We did not impose a timeout for these experiments.

### B.3 BEHAVIOR Domain Experimental Setting

Next, we describe the precise details of our lifelong learning evaluation on BEHAVIOR problems.

### B.3.1 Domain Descriptions

We considered 10 BEHAVIOR problems using the simulated humanoid: boxing books up for storage, collecting aluminum cans, locking every door, locking every window, organizing file cabinet, polishing furniture, putting leftovers away, re-shelving library books, throwing away leftovers, and unpacking suitcase. Following prior work to adapt BEHAVIOR domains to the TAMP setting [23], only actions with PLACEONTOP(object, surface), PLACEINSIDE(object, surface), PLACEUNDER(object, surface), PLACENEXTTO(object, target, surface), NAVIGATE-TO(object), and GRASP(object, surface) controllers were implemented at the continuous level,

while other actions (e.g., CLEANDUSTY or OPEN) were implemented only at the abstract level and assumed to always succeed if their abstract preconditions held.

### B.3.2  Auxiliary Geometric Signals

The auxiliary signals that we used to assess each sampler's reliability were:

- NAVIGATETO: sine and cosine of the robot's yaw; distance to target; nearest point on the object's bounding box (in the object's frame); distance to the nearest point on the object's bounding box; and robot position (in the object's frame).

- GRASP: sine and cosine of the Euler angles of the robot's gripper; distance of the gripper to the target and the surface; distance of the gripper to the nearest point on the target's and surface's bounding boxes; position, and sine and cosine of the Euler angles of the gripper's pose (in the target's and surface's frames).

- PLACE $\cdots$ : distance from hand and object to surface; nearest points from hand and object to surface's bounding box (in the surface's frame); nearest point from hand to object's bounding box (in the object's frame); distances to these nearest points; positions, and sines and cosines of Euler angles of the gripper's and object's poses (in the surface's frame). For PLACENEXTTO, we additionally computed the relevant distances, nearest points, and relative coordinates with respect to the target object.

Like in the 2D domains, these signals measure only intended effects, but have no means to effectively measure if those effects are attained (e.g., due to collisions with unforeseen objects).

### B.3.3  Constructing the Mixture Distribution

In BEHAVIOR domains, we only considered the trained specialized and generic samplers as mixture components, since computing the uniform sampler's error like in the 2D case would have required precomputing the error of random predictors via simulation, which was prohibitively expensive for BEHAVIOR. In consequence, we used the root mean square error directly (without normalization) to weight the two mixture components.

### B.3.4  Lifelong Training Details

In the lifelong setting, we only used the learned samplers for exploration when both generic and specialized samplers had been trained. Whenever a new object type was encountered, hand-crafted samplers were used. At the start of the robot's lifetime, all samplers were initialized to hand-crafted distributions from prior work [23]—note that, for BEHAVIOR domains, a uniform distribution, like we used in the 2D domains, would never complete problems within any reasonable time limit.

### B.3.5  Evaluation Protocols

We repeated the BEHAVIOR experiments over four trials with varying random seeds, which controlled both the order of BEHAVIOR problem families and the sampled problems within each family. We trained the agent sequentially on all ten families in each trial. We presented the robot with $96$ problems of each family in sequence, and updated models every $48$ problems.

We again used Fast-Downward as the skeleton generator with $N = 1$. We set the sample bound to $B = 1,000$, with up to $M = 10$ samples at each state before backtracking. We did not use a timeout for these experiments.

## C  Complete Results of Learning from Fixed Data on 2D Domains

Tables C.1 and C.2 show the mean and standard error across trials of the number of samples and number of solved problems in the experiments of Section 5.2 in the main paper.

Table C.1: Average $\pm$ standard error of the number of samples needed to solve problems in 2D domains after training diffusion models from offline data—accompanying table for Figure 4 in the main paper. Variance across trials is very small, demonstrating the statistical significance of our results.

| Sampler choice | 50 problems | 500 problems | 5,000 problems | 50,000 problems |
|---|---|---|---|---|
| Specialized | $3970.48_{\pm 89.04}$ | $2071.54_{\pm 43.93}$ | $1541.26_{\pm 46.41}$ | $1161.38_{\pm 41.87}$ |
| Generic | $2538.38_{\pm 64.12}$ | $1830.37_{\pm 48.14}$ | $1454.18_{\pm 49.16}$ | $1129.52_{\pm 40.06}$ |
| Mixture | $1904.38_{\pm 57.82}$ | $1469.85_{\pm 52.96}$ | $1302.50_{\pm 42.83}$ | $1097.14_{\pm 40.53}$ |
| CD generic | $2324.82_{\pm 48.25}$ | $1720.13_{\pm 32.23}$ | $1376.16_{\pm 54.48}$ | $1333.63_{\pm 47.58}$ |
| CD mixture | $1676.47_{\pm 36.36}$ | $1426.21_{\pm 32.79}$ | $1297.18_{\pm 32.88}$ | $1134.70_{\pm 47.31}$ |
| Uniform | $3063.76_{\pm 60.13}$ | $3063.76_{\pm 60.13}$ | $3063.76_{\pm 60.13}$ | $3063.76_{\pm 60.13}$ |
| NSRTs [4] | $8472.66_{\pm 59.78}$ | $2674.26_{\pm 42.31}$ | $2024.20_{\pm 55.44}$ | $1927.29_{\pm 54.28}$ |

Table C.2: Average $\pm$ standard error of the number of solved problems in 2D domains after training diffusion models from offline data—accompanying table for Figure 4 in the main paper. Variance across trials is very small, demonstrating the statistical significance of our results.

| Sampler choice | 50 problems | 500 problems | 5,000 problems | 50,000 problems |
|---|---|---|---|---|
| Specialized | $76.96_{\pm 0.98}$ | $93.44_{\pm 0.48}$ | $95.32_{\pm 0.47}$ | $96.44_{\pm 0.36}$ |
| Generic | $89.84_{\pm 0.55}$ | $94.32_{\pm 0.49}$ | $95.36_{\pm 0.43}$ | $96.28_{\pm 0.32}$ |
| Mixture | $94.28_{\pm 0.55}$ | $96.32_{\pm 0.52}$ | $95.60_{\pm 0.31}$ | $96.44_{\pm 0.39}$ |
| CD generic | $90.64_{\pm 0.33}$ | $94.48_{\pm 0.33}$ | $95.84_{\pm 0.36}$ | $95.60_{\pm 0.46}$ |
| CD mixture | $94.88_{\pm 0.32}$ | $95.56_{\pm 0.33}$ | $95.72_{\pm 0.32}$ | $96.20_{\pm 0.41}$ |
| Uniform | $92.76_{\pm 0.54}$ | $92.76_{\pm 0.54}$ | $92.76_{\pm 0.54}$ | $92.76_{\pm 0.54}$ |
| NSRTs [5] | $23.16_{\pm 0.89}$ | $89.00_{\pm 0.38}$ | $93.72_{\pm 0.40}$ | $94.32_{\pm 0.49}$ |

# D   Additional Experiments

In this section, we present ablations and additional experiments to those presented in Section 5 in the main paper.

## D.1   Evaluating the Mixture Weight Construction

While the results of our main experiments strongly support the choice of using mixture distributions for generating samples for TAMP, we were interested in more clearly understanding the choices of how to construct those mixture distributions. For this purpose, we implemented and evaluated five alternative strategies for constructing the mixture distribution from our nested models:

- **Distance**   Our first alternative mixture still follows the process of training auxiliary models but, unlike our main implementation, uses only a single auxiliary variable: distance to target. This allows us to verify whether a collection of auxiliary signals is necessary, or a single one may suffice.

- **Reconstruction**   We similarly create an auxiliary model for directly reconstructing the state features $x$. With this, we check the usefulness of including the action parameters $\phi$ in the auxiliary tasks.

- **Uniform**   This strategy simply uses a uniform mixture distribution. The purpose of evaluating this technique is to check whether all the gains of mixture distributions come from the mere fact of using a mixture, or the weighting plays an important role.

- **Proportional**   This *cheating* method observes the outcomes of the uniform mixture over all test problems, and computes the mixture weights as the proportion of successful samples that were drawn from each mixture component. We include this strategy to check whether there may exist some fixed choice of mixture weights that works *across all states*.

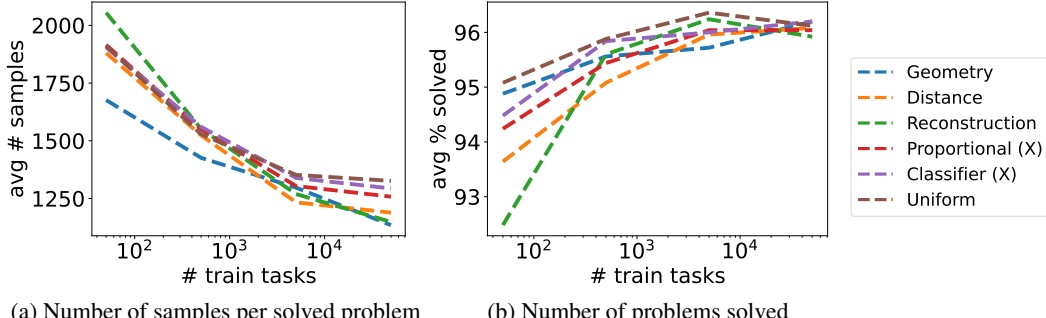

(a) Number of samples per solved problem     (b) Number of problems solved

Figure D.3: Alternative choices of the mixture strategy to generate samples for TAMP in the 2D domains. Our geometric prediction method is most effective in the low-data setting.

- **Classifier** This additional *cheating* method also observes the outcomes of the uniform mixture and trains a classifier to generate the mixture weights. This enables us to study whether it may be possible to train a model to directly choose which sampler to use, as an alternative to using auxiliary tasks as an assessment of reliability.

The results across the five 2D domains are shown in Figure D.3. While all mixture choices perform well, in the low-data regime, which is crucial in lifelong settings, our geometric predictions lead to the highest efficiency across all mixture choices. Notably, the uniform mixture solves the largest fraction of problems. This is expected, given that our mixtures include the uniform sampler over the action space, which is guaranteed to eventually find a successful sample; only the uniform mixture ensures that this sampler is used sufficiently often to guarantee solving most problems (albeit less efficiently than other samplers). The reconstruction error performs worst of all in the low-data setting, but eventually matches the performance of our geometry-prediction implementation; this validates the importance of including the action parameters $\phi$ as part of the auxiliary signal computation. Neither of the strategies that cheat is especially strong, indicating that 1) fixed mixture weights are not sufficiently flexible and 2) directly predicting which sampler to use, given the state, is difficult.

## D.2 Replay Training Matches Full Retraining in Lifelong Setting on 2D Domains

To validate our use of balanced replay instead of full retraining, requiring $10\times$ fewer training epochs due to starting from previously trained models, we compared the performance of the two methods on the lifelong sequence of 2D domains from Section 5.3 in the main paper. Results in Figure D.4 demonstrate that the two methods perform equivalently.

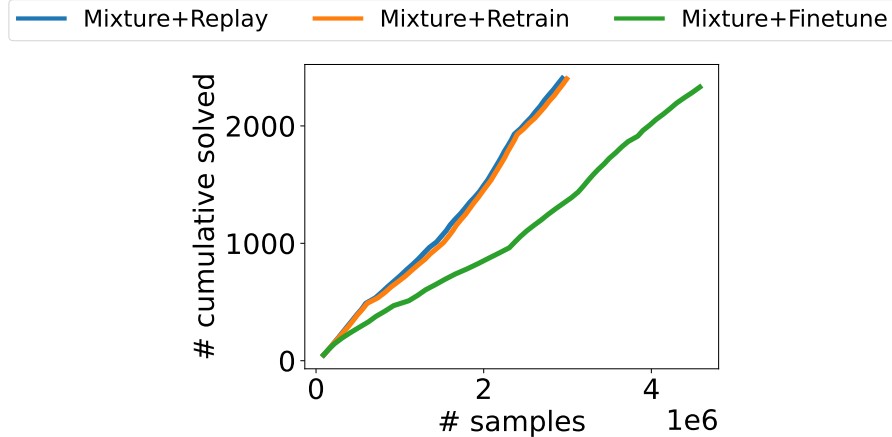

Figure D.4: Comparison of retraining vs replay on the lifelong learning evaluation in 2D domains. Replay (which is more efficient) matches the performance of retraining over the sequence of problems.

