# OpenReview forum: "Embodied Lifelong Learning for Task and Motion Planning"
_robot-learning.org/CoRL/2023/Conference — CoRL 2023 Poster_

### Official Review · Reviewer_8dBQ · 2023-07-17

**Confidence:** 4
**Originality:** Very Good
**Technical Quality:** Fair
**Clarity Of Presentation:** Good
**Impact:** 4

**Recommendation:**

Weak Accept: I recommend accepting the paper, but will not argue for my recommendation if the majority of other reviewers have a different opinion.

**Review:**

Strengths:
- Propose the formulation of lifelong learning in TAMP setting.

Weakness:

Overview:
The major shortcoming of this paper is the definition of “tasks.” I think the definitions are vague and the tasks in each domain have very concrete explanations. Is one task in the 2D domain a sampled initialization + a goal to achieve in the domain? Are the “tasks” equivalent to the “problems” defined in the formulation? Or are “tasks” a subset of “problems”? Without a clear definition, it is very hard to evaluate the applicability of the proposed setting of lifelong learning TAMP, and if it is a reasonable thing to ignore previous tasks.

Detailed points:
- While it is commonly recognized that in the real world, a person / a robot does not encounter the exactly same situation twice, as for research, not evaluating the previous problem seems strange. I think evaluating prior problems is the best way to quantify how much forgetting an agent experienced and see the correlation between knowledge forgetting and the agents’ performance on new tasks.
- Ln74-76: I disagree with this statement. Sometimes forgetting past knowledge is good — prior knowledge might be conflicting in performing new tasks. Either the authors formulate problems and define the scope that forgetting knowledge will definitely impact future learning negatively, or this statement is not right to be claimed.
- The formulation in Section 2.2 is not clear. It is very confusing to see an abstraction action as an element from the space of \mathcal{O}. At the same time, its notation is not o and its corresponding controller has object-based parameters o. Please make the notations consistent, or at least introduce symbols that would not introduce ambiguities.
- More importantly, why is “bilevel planning” mentioned here? Is bilevel planning just task and motion planning in this context? If so, can you provide a more detailed explanation before introducing bilevel planning?
- Figure 2 (Same comments apply to Figure 3): I cannot see the differences in distributions between the “observed” and “learned”. Maybe highlight the differences in Figure 2 so that it’s obvious what the better samples the diffusion model can give.
- 2 /3 of the points in the limitations have nothing to do with the lifelong learning setting in TAMP. I think those two points apply to other classical TAMP works as well. I think the authors need to mention more of limitations that are more specific to lifelong learning in TAMP.

**Quality Of The Limitations Section:**

Limitations are addressed clearly

**Questions For Rebuttal:**

- Authors defined an action $a$ as the result of applying a controller (Ln98). Is it equivalent to a (sub)goal state? If so, why is it defined as an action instead of a state?
- Please address the comments on the weakness, mainly about the symbols in formulation, figure 2, limitations, and all the ambiguities in writing.
- Is the diffusion model a standard model for sampling parameters in TAMP? I believe it definitely needs to be compared against some existing samplers, either a hand-crafted one or an active sampling strategy.

**Robotics Focus:**

Relevant but unlikely to deploy to hardware in near future

**Summary Of Paper:**

This paper presents a new formulation of Task-and-Motion-Planning — a lifelong learning setting of TAMP. This paper specifically focuses on learning the models for samplers of continuous parameters in TAMP problems.


**Summary Of Recommendation:**

While I like the concept of lifelong learning in TAMP settings, the manuscript lacks the clarity of the problem formulation, and some of the experiments are confusing and unclear. If the authors can significantly improve the paper clarity by addressing my questions, I will definitely increase the score if all the questions are addressed.

After rebuttal: I am very borderline on this paper submission. I don't think it's completely convincing to me since they have an important experiment missing that quantifies forgetting explicitly, but other than that, the authors have addressed all my concerns. I am neither up for nor against its acceptance.

---

### Official Review · Reviewer_VjVf · 2023-07-17

**Confidence:** 4
**Originality:** Fair
**Technical Quality:** Good
**Clarity Of Presentation:** Good
**Impact:** 2

**Recommendation:**

Weak Accept: I recommend accepting the paper, but will not argue for my recommendation if the majority of other reviewers have a different opinion.

**Review:**

The paper is well-motivated, and the problem is clearly defined. The following are the main strengths of this paper:
1. Using a learned sampler, the authors show improvement in reducing the number of samples and improving task performance.
2. The online training of models efficiently is a challenging problem for the robotics community, and the paper addresses this challenge for diffusion models in this work.

As the authors mention in the paper (lines 106-109), the simulation step that checks if the sampled point can solve the underlying task is quite expensive. If the authors could provide the timing results for task completion between the learned and uniform sampling methods, it would help validate their claim, especially since it is known that diffusion models fall short in sampling speeds.

Here are a few more suggestions for improving the clarity of the paper:
1. Although the authors tried explaining their method in Sections 2, 3, and 4, a figure outlining their approach, like in Fig 1 in [1], would make it easier for readers to follow their method.
3. A video showing the high-level plan for the BEHAVIOR dataset would help understand the complex distributions learned by the diffusion model.

[1] R. Chitnis, T. Silver, J. B. Tenenbaum, T. Lozano-Perez, and L. P. Kaelbling. Learning neuro-symbolic relational transition models for bilevel planning. In 2022 IEEE/RSJ International Conference on Intelligent Robots and Systems (IROS-22), pages 4166–4173.

**Quality Of The Limitations Section:**

Limitations are addressed clearly

**Questions For Rebuttal:**

1. What do skeleton_gen.next() and 's' mean in Algorithm 1?
2. What does 'o' represent in line 94? Is it different from the `o' used to represent objects in line 87?
3. In Section 5.2, when training the model under the mixture dataset, when N=50, how many samples from each task are used?
4. How many tasks would it solve if the uniform sampler is used in the lifelong environment?

**Robotics Focus:**

Highly relevant to robotics but no hardware experiments

**Summary Of Paper:**

This paper introduces a lifelong learning framework that uses generative models to sample grasps, poses, and paths for a high-level plan. To generate these samples, the authors propose a mixture model that weighs between an object-centric and generalized model for the resulting distribution. Each distribution is modeled using a diffusion model for its ability to capture complex distributions. The authors evaluate their learning strategy in an offline 2D environment and show that a diffusion model that shares data over multiple tasks performs better than models that use specialized datasets. The authors also demonstrate that the proposed framework can learn and improve task performance in a 2D environment and the BEHAVIOR dataset.

**Summary Of Recommendation:**

The proposed method does reduce the number of samples and improves task performance. But assessing this work's impact is difficult because no experiments or data compare the time required to sample using diffusion models and the time needed to test points and find a valid solution. If the authors can add experiments to clarify this point, it would help better understand the impact of their work.

---

### Official Review · Reviewer_2eBC · 2023-07-20

**Confidence:** 4
**Originality:** Very Good
**Technical Quality:** Very Good
**Clarity Of Presentation:** Very Good
**Impact:** 4

**Recommendation:**

Weak Accept: I recommend accepting the paper, but will not argue for my recommendation if the majority of other reviewers have a different opinion.

**Review:**

The paper is well written, authors motivate the work and provide relation to some other relevant works in the field. The problem is very useful for the employment of TAMP solutions. However, several open questions and a few stylistic recommendations are stated.

The presented method is not explained in enough detail. It is basically kept on a high level and an appendix is very much needed to understand the approach.

The readability could be improved. For example, some of the wording is too strong. e.g., “robot will have to use” → “robot could use”. And some wordings are not directly understandable by non-native English speakers or are not academic language (e.g., conducive, grow stronger, etc.)

Demonstration domains are very simplistic with short-horizon planning and it is not clear what is the distribution of the training examples and how much they differ in the lifelong operation. A more fair baseline for comparison could be provided.

In the specialized models, one hot encoding is used, which is very limiting for lifelong operation when novel object types can appear. On the other hand, the generic sampler is used on a problem of limited size and therefore the scalability to complex scenarios is not visible.

**Quality Of The Limitations Section:**

Additional details required

**Questions For Rebuttal:**

Improving readability would be a minor point as stated in the review.

Although it is clear for readers with TAMP background, a more appropriate introduction to samplers would be useful.

My main concern is in the demonstration domains and learning experiments. It is not fully clear how experiments are done, as there are no training and testing scenarios it is not clear if it is somehow overfitting to the scenario
.
Besides that, the contribution of the Diffusion approach is also not visible.

Comparison with some other learning-based samplers would be useful. (i.e. LOFT [1])

[1] Silver, Tom, et al. "Learning symbolic operators for task and motion planning." 2021 IEEE/RSJ International Conference on Intelligent Robots and Systems (IROS). IEEE, 2021.

**Robotics Focus:**

Highly relevant to robotics but no hardware experiments

**Summary Of Paper:**

The paper is dealing with the problem of lifelong learning in task and motion planning. The main approach relies on learning samplers for faster planning in subsequent ongoing operations.
Authors utilize Diffusion Models to learn samplers. The authors also slightly modify the lifelong learning setting and make it more strict that the agent is never evaluated on any of the previous problems.

**Summary Of Recommendation:**

I believe the work is interesting, tackling an interesting problem, very useful for the employment of TAMP and uses an appropriate approach. However, I believe it is still not mature enough and relies on the use of Diffusion models without appropriate demonstration and comparison to baselines.

---

### Official Review · Reviewer_h46E · 2023-07-27

**Confidence:** 4
**Originality:** Good
**Technical Quality:** Good
**Clarity Of Presentation:** Good
**Impact:** 4

**Recommendation:**

Weak Accept: I recommend accepting the paper, but will not argue for my recommendation if the majority of other reviewers have a different opinion.

**Review:**

**Strengths**
* The paper addresses a relevant and open problem in robot learning — how can robots efficiently and continually learn to solve new tasks? — and the method is well motivated.
* The proposed evaluation metric — comparing methods based on cumulative performance on a sequence of tasks that never exactly repeat and doing away with train/test splits — is original and interesting. It is also well motivated. I think it is highly relevant to a long term goal of robot learning of deploying robots in homes and hope to see this adopted broadly.
   * I especially appreciated the discussion in lines 70 - 81 about why avoiding forgetting is still important in the proposed evaluating setting.
* The proposed method is interesting and to my knowledge original, especially making use of a mixture of learned samplers using diffusion models and learning auxiliary predictors to select the weights between the different models. Both of these choices are practical and interesting and I especially appreciated the implementation discussion in section 3.
* The set of experiments is mostly well chosen and they shed insight into why the proposed method works.
   * Sufficient data is provided to support the key method design choices through ablation studies. For example, assessing the different strategies for learning samplers (Fig 5), different approaches for incorporating new data (finetune, replay, retrain (Fig 4, Fig B.3), and for weighting different samplers (Fig A.2)
   * I enjoyed the discussion in 5.2 and 5.3 - it was insightful.
   * There are a few additional comparisons that I would like to see (please see below).
* The paper is mostly clear and well written, and sufficient detail is mostly provided about the method to re-implement it. However there are a few places where the clarity and detail provided could be improved (see below)

**Weaknesses**
* There are no experiments on a real robot and no discussion about why this is the case or whether the method could be deployed on a real robot in the near future.
* Whilst the components of the method are well ablated, the proposed method is not compared with any other approaches for solving the same problems. There is also no discussion about why no such comparison is appropriate.
   * This makes it difficult to assess when to use this method compared to already existing approaches.
   * [5] seems like a viable candidate. Please could the authors include a [5] as a baseline method or include a discussion as to why this is not appropriate.
* The method assumes that the robot has access to a sound and probabilistically complete planner.
   * I am a little concerned about how strong an assumption this is and thus how broadly applicable the proposed method is. It would be helpful to include a discussion of how stringent this assumption is.
* There are a few areas where the clarity and level of detail provided could be improved
   * Abstract: “Our method exhibits substantial improvements in planning success” - compared with what? What are the baselines?
   * Introduction, lines 29-31: Can you provide evidence to support the claim that finding continuous parameters that guarantee the success of a high level plan is the most difficult aspect of TAMP?
   * Section 2.2: line 93: O does not appear to be defined.
   * The 1-step and N-step labeling in figures 2 and 3 is a little confusing since it is not referred to in the text. How do they relate to the two data sets collected for each domain (valid action, task success actions)? Additionally, observed and learned look very similar. Presumably this is intended and it would be helpful if the similarities between the two could be discussed.
   * Fig 4: Is mixture+replay a typo? If not, why is only replay combined with mixture and retrained combined with specialized/generic?
       * [minor] It would also be nice to see uniform sampling included as a baseline in Fig 4.
   * Line 253: I appreciate that evaluations were performed over 10 trials however it would be useful if the variance in results could be reported. Also, presumably the reported results in Fig 5 are the average. Please could this be clarified.
   * No detail is provided on the simulator in which the environments are implemented. To improve the reproducibility of this work, at least some detail about this would be useful.
   * Related work: “Our focus is on enabling a robot system to become increasingly capable over time via compositionality” This is the first time in the paper that compositionality is mentioned. It is not clear how the proposed method leads to compositional behavior.
* The evidence provided in section 5.4 does not appear sufficient to establish the benefits of the proposed approach in the BEHAVIOR domain.
   * No comparison with the uniform baseline or the specialized or generic models is provided in Fig 6 (in contrast it is in Fig 4).
   * Line 300: Why are the hand-crafted samplers expected to be a strong baseline? Please could the authors provide a brief discussion on this.

**Quality Of The Limitations Section:**

Additional details required

**Questions For Rebuttal:**

In addition to addressing the points and questions raised in the previous section, please could the authors address the questions below:
* Line 213: What data does Z_new consist of? Is it just the samples that led to task success?
* Lines 274-276: Was there a sampling budget or was the sampler permitted to continue sampling until the problem was solved?

**Robotics Focus:**

Relevant but unlikely to deploy to hardware in near future

**Summary Of Paper:**

This paper proposed a method for lifelong learning of continuously new robot task and motion planning (TAMP) problems. This work focuses on learning continuous parameters of a controller (e.g. grasps, poses, paths) that guarantee the success of a (given) discrete high level plan. To achieve this, the proposed method learns a weighted mixture of diffusion models that outputs continuous parameters given the environment state and a parameterized controller. The method is evaluated on a number of 2D domains and 10 tasks from the BEHAVIOR benchmark and the different design choices are assessed experimentally.

**Summary Of Recommendation:**

An interesting and mostly well written paper which contains a number of original ideas that I think would be interesting to the research community. The proposed method is an original approach to lifelong robot learning, as is the proposed evaluation protocol. I can imagine it inspiring further research. The technical assessment of the method is quite good but there are a few areas where it could be improved. In particular, the lack of a baseline makes it difficult to assess when to use this method compared to already existing approaches. Further the method is only assessed in simulation with no discussion about a path to deployment on hardware.

On balance this paper currently stands at a weak reject for me but I would be happy to revise my recommendation if the authors address my comments and questions.

=====

Following the rebuttal period and discussion I have revised my recommendation to **weak accept**

---

### Author Response · Authors · 2023-08-11
**General response to all reviewers**

We thank the reviewers for their valuable comments. We believe that the feedback and suggestions provided here have collectively improved our manuscript substantially. We hope to engage all reviewers in discussion over the coming days.

We provide individual responses to each reviewer, and here we provide a brief summary of the main points in our responses + any major changes to our manuscript. Any additions to the PDF are highlighted in **BLUE**.
1. **Comparison to baseline:** In our revised manuscript, we have included [4] (previously [5]) as a baseline method in Fig. 4 (sampler learning from offline data), as suggested by Reviewer h46E. To make comparisons fair, we provided [4] with the known operators, and used it only to learn the samplers---[4] originally learns samplers and operators, but using learned operators would have put [4] at an unfair disadvantage. In summary, [4] is incapable of learning a useful distribution for planning with N=50 tasks (its performance is substantially worse than that of the uniform sampler). Even after N=50,000 tasks, the learned samplers still perform worse than all our diffusion-based baselines. This is expected: [4] makes strong assumptions about the shape of the distribution of promising candidate actions---namely, that it can be represented via rejection sampling over a Gaussian distribution. In contrast, diffusion models can represent distributions of complex, unknown shapes.
2. **Uniform baseline in lifelong learning:**  We have also added the comparison to the uniform sampler in Fig. 5. Interestingly, the uniform distribution is better than any of the distributions learned with approaches other than mixture+replay, corroborating the need for a more intelligent approach to sharing (or not) data across samplers (like our mixture approach).
3. **Clarity of notation:** To clarify, in our original manuscript we used $\mathcal{O}$ for the set of operators, and 'o' for objects. We realize how this may have been confusing, and the confusion was likely increased by not explicitly introducing a (lowercase) symbol for a single operator. We have made revisions to our notation in the updated manuscript. Namely:
    - We now use 'e' (as in entity) for objects
    - We use 'o' and '$\mathcal{O}$' for operators and the set of operators
    - We clarify that an abstract action is a tuple (operator, controller), which is why we use a different symbol ('a', instead of 'o')
4. **More extensive experiments on BEHAVIOR:** After submission of our original manuscript, we ran our BEHAVIOR experiments for longer. Originally, we had run 6 behavior problem "families" (drawn randomly from a set of 10 in each trial) in sequence, each over 96 problems. We have now included results over all 10 behavior families in sequence, each with 240 problems. This substantially longer lifelong deployment resulted in noticeably larger differences between our learned samplers and the hand-crafted samplers.

---

### Decision · Program_Chairs · 2023-08-30

**Decision:**

Accept (Poster)

**Comment:**

The paper addresses the problem of lifelong task and motion planning with diffusion models as samplers for grasps, poses and paths as they can capture complex distributions. Experiments demonstrate the benefits of the approach in an offline 2D environment and with the BEHAVIOR dataset.

Reviewers appreciated the novelty of the work but were initially concerned about the lack of comparisons to other methods and lack of experiments with the real robot. The authors have addressed most of the concerns during the rebuttal phase, particularly with the addition of comparisons in simulation. Authors are encouraged to provide videos demonstrating the sampling strategy and possible demonstrations on the real robot.